# Delays in seeking treatment for fever in children under five years of age in Nigeria: Evidence from the National Demographic Health Survey

**Anayochukwu E. Anyasodor**[1]*, **Kedir Y. Ahmed**[1], **Uchechukwu L. Osuagwu**[2,3], **Nnamdi C. Mgbemena**[4], **Bernd H. Kalinna**[1], **Subash Thapa**[1], **Shakeel Mahmood**[1], **Allen G. Ross**[1]

1 Rural Health Research Institute, Charles Sturt University, Orange, NSW, Australia, 2 Translational Health Research Institute, Western Sydney University, Campbelltown, Penrith, NSW, Australia, 3 Bathurst Rural Clinical School (BRCS), School of Medicine, Western Sydney University, Bathurst, NSW, Australia, 4 School of Allied Health, Exercise and Sports Sciences, Charles Sturt University, Orange, NSW, Australia

* aanyasodor@csu.edu.au

**Data Availability Statement:** The data used for this study are available from The DHS Program. Users can register on the DHS Program website

## Abstract

### Background

In countries with high child mortality rates, such as Nigeria, early intervention for common childhood illnesses (e.g., pneumonia and malaria) is essential for improving clinical outcomes. The timely reporting and treatment of fever is therefore critical in making a differential diagnosis and choosing an appropriate course of treatment. The present study aimed to investigate the prevalence and major risk factors associated with delays in seeking treatment for fever in children under five years of age in Nigeria.

### Methods

This study used a total weighted sample of 7,466 children under five years of age from the 2018 National Nigerian Demographic and Health Survey. Multivariable binary logistic regression modelling was used to investigate the association between predisposing, enabling, need, health service and community level factors, and delay in treatment-seeking for fever.

### Results

We report the delays in seeking treatment for childhood fever that was reported by mothers in the last two weeks prior to the national survey. The prevalence for delayed treatment was 62.1% (95% confidence interval [CI]: 60.1%, 64.1%). Our findings showed that there were fewer delays in seeking treatment in children aged 24–59 months (adjusted odds ratio [aOR] = 0.79, 95% CI: 0.68, 0.93), among mothers who were formally employed (aOR = 0.84; 95% CI: 0.73, 0.96), regularly attended antenatal services (aOR = 0.76, 95%CI: 0.66, 0.88), and for those who resided in wealthier households (aOR = 0.71; 95% CI: 0.56, 0.89). Children whose mothers resided in the North-West geopolitical zone of Nigeria were less

(https://dhsprogram.com/). Once registered, interested researchers can request access to the datasets. The 2018 Nigeria DHS data are available for download at the following link: https://dhsprogram.com/data/dataset/Nigeria_Standard-DHS_2018.cfm?flag=0. The authors confirm they had no special access privileges.

**Funding:** The authors received no specific funding for this work.

**Competing interests:** The authors have declared that no competing interests exist.

likely to delay seeking treatment for fever (aOR = 0.55; 95% CI: 0.42, 0.73). However, mothers who had an unwanted pregnancy had a higher odds of delaying treatment for childhood fever (aOR = 1.58; 95% CI: 1.05, 2.39).

## Conclusion

There were significant delays in seeking treatment for childhood fever in poorer homes found in geopolitically unstable zones of Nigeria. Mothers who were poor, unemployed, and with younger children (<12 months) often delayed seeking treatment for their febrile child. Future health promotion strategies and microenterprise schemes should target both rural and urban mothers residing in poor households. Children under 12 months of age should be a priority.

## Introduction

Despite the global endorsement of the Sustainable Development Goals (SDG Target 3.2) to end preventable deaths of newborns and children under-five years worldwide, approximately five million children still die before their fifth birthday annually [1]. Sub-Saharan Africa (SSA) and Southern Asia accounted for more than 80% of these deaths [2], West and Central Africa contributed the highest regional under-five mortality rate (U5MR) of 94.7 deaths per 1000 livebirths [1]. Nigeria currently ranks highest as a country, contributing the largest number of global U5MR of 110 deaths per 1000 live births [3]. This has enormous socioeconomic consequences, and impedes progress toward achieving SDGs in Nigeria [4].

Childhood fever, defined as a temperature at or above 100˚F or 38˚C, is a common clinical symptom of common childhood illnesses such as pneumonia, malaria and measles, and these illnesses are responsible for a significant number of childhood deaths [5–8]. However, effective interventions for these illnesses depend on several factors, including prompt treatment-seeking behaviour at healthcare institutions [9,10]. The time of treatment-seeking is crucial in the effective management of diseases, including early diagnosis, which can prevent exacerbation of the condition [11,12]. Delayed treatment-seeking behaviour can result in complications and increased mortality.

Previously published studies in SSA have shown several factors associated with treatment-seeking behaviour of mothers for childhood fever. In SSA, treatment-seeking behaviour of mothers/caregivers could be influenced by socio-demographic and economic factors such as a child's gender and age [13,14], maternal age [6,15], education level [13], occupation of parent (s)/caregivers [14,15], cultural beliefs [16], availability and location of healthcare facilities [6,17], and family relationships and intra-household decision-making process among parents [18]. Therefore, understanding the above-mentioned determinants of treatment delay will lead to strengthening interventions that will prompt improvements in early diagnosis and treatment of childhood diseases, especially in Nigeria.

Appreciating country-specific attitudes toward treatment-seeking for fever, will substantially strengthen efforts at establishing long-term intervention. There is a lack of nationwide reports in Nigeria to rationalise determinants of delayed treatment-seeking behaviour of caregivers for feverish under-five children, and this study aims to make such contribution. Andersen's Behavioural Model of Healthcare Use provides an extensive list of interconnected factors for health care use, but not all are well understood regarding systematic differences among

sub-populations in treatment-seeking for fever. It is expected that the study outcome presents relevant information, to justify interventional programmes to promote early treatment-seeking for fever in under-five children. This study contributes data to rationalise delay in treatment-seeking behaviour in the country.

Addressing the current situation in Nigeria has strong implications, as it aligns with global and local initiatives such as the SDGs, especially Goals 1, 3, 10 and 17 [19] and the Global Action Plan for Prevention and Control of Pneumonia and Diarrhoea [20]. Improving health-care access and promotion of prompt treatment-seeking behaviour are necessary to mitigate child mortality, prevent infectious diseases, alleviate poverty, minimise inequalities and attain sustainable development required to achieve global health. Therefore, the study was aimed at assessing factors associated with delay in treatment-seeking behaviour for fever among children under-five years of age in Nigeria.

## Methods

### Study setting

Nigeria is a multicultural country in West Africa, and shares borders with Niger in the north, Chad in the northeast, Cameroon in the east; Benin Republic in the west. To the south lies the Gulf of Guinea in the Atlantic Ocean [21]. Nigeria is made up of 36 states and the Federal Capital Territory (FCT), Abuja. A total of 774 local government areas (LGAs) are located in the states and FCT. Each LGA is further divided into wards, the lowest administrative units [22,23]. Based on various criteria such as ethnic composition and culture, Nigeria is divided into 6 geo-political zones: South-South, South-East, South-West, North-Central, North-East and North-West [24].

### Data sources

This study utilised data from the 2018 Nigerian Demographic Health Survey (NDHS), which were assessed on 24/10/2022; and authors had no access to information that could identify individual participants during or after data collection. The data were collected by the National Population Commission (NPC) and the National Malaria Elimination Programme (NMEP) of the Federal Ministry of Health, Nigeria to contribute towards the formulation of policy and coordinating population activities in Nigeria.

A stratified, two-stage cluster design was employed in selecting the study participants. In stage one, a total of 1,400 Enumeration Areas (EAs) were selected in each sampling stratum with probability proportional to the size of the EA. The first stage also included conducting a full household census to determine the number of households in each of the selected EAs. In the second stage of selection, a fixed number of 30 households were drawn from each cluster through equal probability systematic sampling, which gave rise to a sample size of approximately 42,000 households [23]. Out of the total of 41,668 households selected, 40,666 were occupied; and 40,427 were interviewed. Of the 42,121 eligible women enrolled in the study, 41,821 responded to the survey (99.3%). For this study, data for 7,466 children aged 0–59 months and born to the responding mothers were included. The detailed methodology for the 2018 NDHS survey has been reported elsewhere [23].

### Outcome variable

The outcome variable for the study was delay in treatment-seeking for fever, which was defined as seeking treatment after 24 hours, from the onset of fever in children under the age of five years. The outcome variable was dichotomised as "*delayed in treatment-seeking for*

*fever*" (1) or "*not delayed in treatment-seeking for fever*" (0). The decision to use a 24-hour threshold for defining the delay in seeking treatment for fever was informed by the DHS statistics guideline [25] and previously published studies [6,26–28]. Note the delay in seeking treatment for childhood fever was reported by mothers in the last two weeks prior to the national survey.

## Study factors

Fig 1 presents the conceptual framework, which is in line with Andersen's Behavioural Model of Healthcare Use [29]. The study factors were categorised into five classes, stemming from environment, population characteristics and health behaviour (Fig 1). These include community level factors, predisposing factors, enabling factors, need factors and health service factors. Community-level factors included place of residence (urban and rural) and geographical region (North Central, North East, North West, South East, South West and South South). Predisposing factors included child age (0–11, 12–23 and 24–59 months), child gender (male or female), mother's age (15–24, 25–34 and 35–49 year), educational level of the mother and father (no formal, primary, and secondary education), whether or not they listened to the radio (no or yes), read magazine (no or yes) and watched television (no or yes).

Enabling factors included household wealth index (poor, middle, rich), mother's and father's occupation (not working, formal employment and informal employment), distance from health facility (big problem and not big problem); mother's involvement in decision making towards seeking medical care (big problem and not big problem), mother's decision

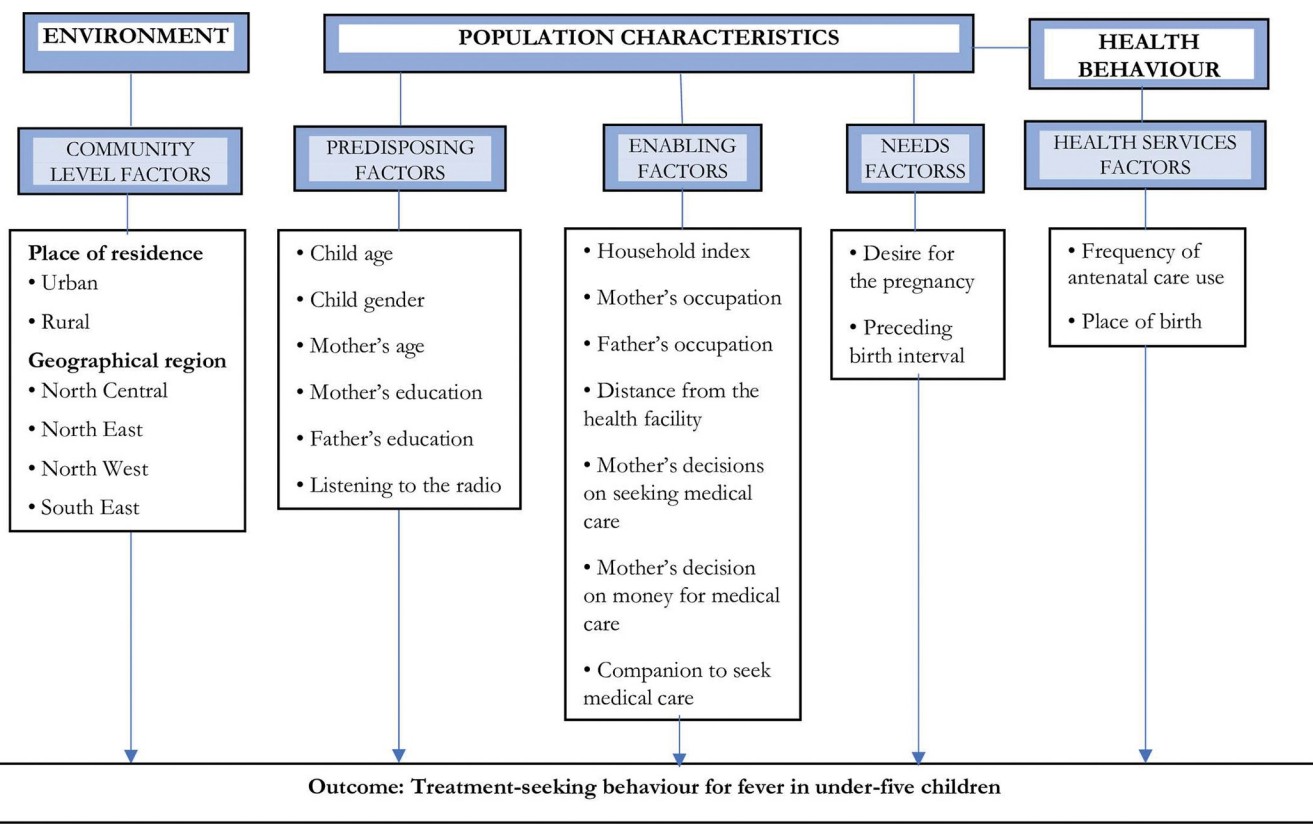

**Fig 1. Modification of the conceptual framework based on Andersen's behavioural model of healthcare use and the corresponding determinants used in our analysis.**

on money for medical care (big problem and not big problem), and companion to seek medical care (big problem and not big problem). Need factors included desire for pregnancy (desired the pregnancy and not desired the pregnancy), and preceding birth interval (no previous birth, <24 months and 24+ months). Health service factors included frequency of antenatal care (none, 1–3 visits and 4+ visits) and place of birth (home and health facility).

## Statistical analysis

The initial analysis involved describing the study participants through calculations of frequencies and percentages related to the study factors, including predisposing, enabling, need, health service and community level factors. This was followed by estimating the prevalence of the outcome variable (delay in treatment-seeking) across these study factors. Multivariable binary logistic regression modelling was used to examine the association between predisposing, enabling, need, health service and community level factors and delay in treatment-seeking for fever. Specifically, predisposing factors were included in the model to examine their relationship with the outcome, while adjusting for enabling, need, health service and community-level factors (Stage 1). A similar strategy was used in models of enabling factors to examine their relationship with the outcome variable, with additional adjustment for predisposing, need, health service and community level factors (Stage 2). The third, fourth, and fifth stages (Stages 3–5) employed similar modelling techniques for need, health service, and community-level factors, respectively.

All statistical analyses were conducted using Stata version 15.2 (StataCorp, College Station, TX, USA) with 'svy' command to adjust for sampling weights, clustering effects and stratification, and the 'melogit' function was used for the modelling. Collinearity was checked using 'variance inflation factor (VIF)' but no significant results were evident in the analyses. Adjusted odds ratios with 95% confidence intervals (CIs) were estimated as the measure of association between study factors and the outcome variable.

## Ethics consideration

The required ethical approvals for the DHS project were reviewed and approved by the Nigerian Health Research Ethics Committee of the Federal Ministry of Health and the ICF Institutional Review Board before the surveys were conducted. A verbal informed consent was sought and obtained from study participants (parents or guardians), prior to data collection while ensuring that their privacy and confidentiality were maintained. The interviewer signed their name, indicating that they read the consent statement to the respondent. Therefore, the respondents were not required to sign their names, as the interviewer had already attested that they followed a proper procedure. The DHS dataset is anonymous, and publicly available with no identifiable information about the survey participants. Another ethical approval was not necessary for this study, since it involves the use of secondary data, however, approval was sought from Measure DHS and permission was granted for this use.

## Results

### Characteristics of the study participants

Overall, 56.6% of children were in the age group 25–59 months, and more than half (53.3%) of mothers did not attain a formal education. 54.9% of children resided in poor wealth index households, and 56.7% of mothers were formally employed in professional jobs. Half of the mothers (50.4%) stated that money for seeking medical care was a big issue for them, and the majority desired the current pregnancy (97.3%). Almost half of the mothers (47.4%) had no

antenatal care (ANC) visit, and more than two-third (68.8%) of them delivered at home (Table 1).

## Delay in treatment-seeking for fever

The overall prevalence of delay in treatment-seeking for fever in Nigeria was 62.1% (95% CI: 60.1, 64.1). Table 2 shows the unadjusted and adjusted analysis of factors associated with delay in treatment-seeking for fever. In the unadjusted analysis, children whose parents had some formal education, read, and listened to the radio were less likely to report delay in treatment-seeking for fever. However, after adjusting for the potential confounders, older children aged 24–59 months, born to fathers with at least a secondary education, and those whose families listened to the radio were less likely to delay in seeking treatment for fever, compared to those who did not. The adjusted odds of delay in seeking treatment for fever was lower for children who resided in wealthy households compared to poor households (aOR = 0.71, 95% CI: 0.56, 0.89). Children whose mothers were formally employed were less likely to delay seeking treatment for fever compared to those who did not have jobs (aOR = 0.84, 95% CI: 0.73, 0.96) [Table 2].

Of the enabling factors, children whose mothers were able to spend funds on medical care were less likely to delay seeking treatment for fever, compared to their counterparts (aOR = 0.75, 95% CI: 0.65, 0.87), after adjusting for potential confounders. The odds of delay in seeking treatment for fever among children whose mothers did not desire their pregnancy was consistently higher than those who desired the pregnancy both in the unadjusted (OR = 1.53; 95% CI: 1.05, 2.26) and adjusted (aOR = 1.58; 95% CI: 1.05, 2.39). Of health service factors, frequency of antenatal service (4 visits or more) was significantly associated with a lower likelihood of delay in treatment-seeking for fever in children, opposed to no antenatal visit (aOR = 0.76, 95%CI: 0.66, 0.88). Parents/caregivers of children who resided in North-West region were less likely to delay seeking treatment for fever compared to those who lived in the North-Central region (aOR = 0.55, 95% CI: 0.42, 0.73) [Table 2].

## Discussion

This study investigated the relationship between predisposing, enabling, need, health service, and community-level factors and the delay in seeking treatment for fever among children under the age of five in Nigeria. Our results indicate that women from wealthier households, those formally employed, who made decisions for treatment-seeking in the household, and those residing in the North-West geopolitical zone were less likely to delay in seeking treatment. Mothers who did not desire their pregnancy were more likely to delay seeking treatment for their children. Our study findings suggested that within the Andersen's Behavioural Model of Healthcare Use [29], enabling factors were more relevant than other factors in determining timely treatment-seeking for fever in children. Household wealth status had an impact on treatment-seeking, and this is consistent with previous studies conducted in other countries such as Ethiopia [30], Eastern Uganda [31], India [6], and Zambia [28]. If essential healthcare for children was made free or greatly subsidised, even for private healthcare institutions, the financial barriers to treatment-seeking would be minimised.

Interestingly, the study revealed that unlike household wealth, women's employment status and ability to make monetary decisions within the household influenced timely treatment-seeking, and has been reported in previous studies [6,15]. Evidence shows that empowerment of women in social, economic, and political areas of their lives has been linked to improved healthcare utilisation and treatment-seeking behaviours [32,33]. More specifically, when mothers have decision-making autonomy within the household, they possess the authority to

**Table 1. Characteristics of the study participants and prevalence of delay in treatment-seeking for fever among children under five in Nigeria, 2018 (N = 7466).**

| Variables | n (%) | Delay for fever treatment | | P-value |
|---|---|---|---|---|
| | | Yes, n (%) | No, n (%) | |
| Predisposing factors | | | | |
| Child age (n = 7466) | | | | |
| 0–11 months | 1247 (16.7) | 804 (64.5) | 443 (35.5) | 0.175 |
| 12–23 months | 1990 (26.7) | 1215 (61.0) | 776 (39.0) | |
| 24–59 months | 4228 (56.6) | 2618 (61.9) | 1610 (38.1) | |
| Child gender (n = 7466) | | | | |
| Male | 3702 (49.6) | 2284 (61.7) | 1418 (38.3) | 0.503 |
| Female | 3764 (50.4) | 2354 (62.5) | 1410 (37.5) | |
| Mother's age (n = 7466) | | | | |
| 15–24 years | 1892 (25.3) | 1151 (60.8) | 741 (39.2) | 0.263 |
| 25–34 years | 3675 (49.2) | 2273 (61.8) | 1403 (38.2) | |
| 35–49 years | 1898 (25.4) | 1214 (63.9) | 685 (36.1) | |
| Mother's education (n = 7466) | | | | |
| No formal education | 3982 (53.3) | 2752 (69.1) | 1231 (30.9) | <0.001 |
| Primary education | 1137 (15.2) | 684 (60.1) | 454 (39.9) | |
| Secondary or higher education | 2346 (31.4) | 1202 (51.2) | 1144 (48.8) | |
| Father's education (n = 7336) | | | | |
| No formal education | 3342 (45.6) | 2348 (70.2) | 995 (29.8) | <0.001 |
| Primary education | 1012 (13.8) | 640 (63.3) | 372 (36.7) | |
| Secondary or higher education | 2981 (40.6) | 1576 (52.9) | 1404 (47.1) | |
| Listening to the radio | | | | |
| No | 3722 (49.9) | 2562 (68.9) | 1159 (31.2) | <0.001 |
| Yes | 3744 (50.2) | 2075 (55.4) | 1669 (44.6) | |
| Reading magazine | | | | |
| No | 6786 (90.9) | 4329 (63.8) | 2457 (36.2) | <0.001 |
| Yes | 679 (9.1) | 308 (45.3) | 372 (54.7) | |
| Watching television | | | | |
| No | 4836 (64.8) | 3271 (67.3) | 1565 (32.4) | <0.001 |
| Yes | 2630 (35.2) | 1366 (52.0) | 1263 (48.0) | |
| Enabling factors | | | | |
| Household wealth index | | | | |
| Poor | 4083 (54.7) | 2859 (70.0) | 1224 (30.0) | <0.001 |
| Middle | 1510 (20.2) | 884 (58.5) | 626 (41.5) | |
| Rich | 1873 (25.1) | 895 (47.8) | 978 (52.2) | |
| Mother's occupation (n = 7466) | | | | |
| Not working | 2075 (27.8) | 1371 (66.1) | 704 (33.9) | <0.001 |
| Formal employment | 4236 (56.7) | 2441 (57.6) | 1795 (42.4) | |
| Informal employment | 1155 (15.5) | 825 (71.4) | 330 (28.6) | |
| Father's occupation (n = 4167) | | | | |
| Not working | 198 (4.7) | 141 (71.1) | 57 (28.9) | 0.006 |
| Formal employment | 2684 (64.4) | 1506 (56.1) | 1177 (43.9) | |
| Informal employment | 1286 (30.8) | 703 (54.7) | 582 (45.3) | |
| Distance from the health facility | | | | |
| Big problem | 2227 (29.8) | 1618 (72.7) | 609 (27.4) | <0.001 |
| Not big problem | 5238 (70.2) | 3019 (57.6) | 2219 (42.4) | |
| Mother's decisions on seeking medical care | | | | |

*(Continued)*

**Table 1.** (Continued)

| Variables | n (%) | Delay for fever treatment | | P-value |
|---|---|---|---|---|
| | | Yes, n (%) | No, n (%) | |
| Big problem | 876 (11.7) | 655 (74.8) | 221 (25.2) | <0.001 |
| Not big problem | 6589 (88.3) | 3982 (60.4) | 2607 (39.6) | |
| Mother's decision on money for medical care | | | | |
| Big problem | 3765 (50.4) | 2598 (69.0) | 1167 (31.0) | <0.001 |
| Not big problem | 3700 (49.6) | 2039 (55.1) | 1662 (44.9) | |
| Companion to seek medical care | | | | |
| Big problem | 1205 (16.2) | 878 (72.9) | 327 (27.1) | <0.001 |
| Not big problem | 6260 (83.9) | 3759 (60.0) | 2502 (40.0) | |
| Need factors | | | | |
| Desire for the pregnancy | | | | |
| Desired the pregnancy | 7262 (97.3) | 4492 (61.9) | 2770 (38.1) | 0.028 |
| Not desired the pregnancy | 204 (2.7) | 146 (71.4) | 58 (28.7) | |
| Preceding birth interval | | | | |
| No previous birth | 1297 (17.4) | 744 (57.3) | 554 (42.7) | 0.003 |
| <24 months | 1404 (18.8) | 1978 (61.2) | 1256 (38.8) | |
| 24+ months | 4758 (63.8) | 1916 (65.3) | 1019 (34.7) | |
| Health service factors | | | | |
| Frequency of antenatal care use | | | | |
| None | 3522 (47.4) | 2368 (67.2) | 1154 (32.8) | <0.001 |
| 1–3 visits | 1067 (14.4) | 703 (65.9) | 363 (34.1) | |
| 4+ visits | 2836 (38.2) | 1551 (54.7) | 1285 (45.3) | |
| Place of birth | | | | |
| Home | 5134 (68.8) | 3399 (66.2) | 1735 (33.8) | <0.001 |
| Health facility | 2332 (31.2) | 1238 (53.1) | 1094 (46.9) | |
| Community level factors | | | | |
| Place of residence | | | | |
| Urban | 2269 (30.4) | 1164 (51.3) | 1105 (48.7) | <0.001 |
| Rural | 5197 (69.6) | 3473 (66.8) | 1723 (33.2) | |
| Geographical region | | | | |
| North Central | 754 (10.1) | 529 (70.2) | 225 (29.9) | <0.001 |
| North East | 1959 (26.2) | 1353(69.1) | 606 (30.9) | |
| North West | 3039 (40.7) | 1821 (59.9) | 1218 (40.1) | |
| South East | 643 (8.6) | 346 (53.8) | 298 (46.2) | |
| South West | 710 (9.5) | 383 (53.9) | 328 (46.1) | |
| South South | 361 (4.8) | 206 (57.0) | 155 (43.0) | |

prioritise their children's health, and promptly seek medical care when needed [32,34]. Empowered women will have better access to resources and information, enabling them to overcome financial barriers and seek timely healthcare for their children [33]. Given this fact, microenterprise strategies that benefit women in general and provision of incentives to women for their healthcare needs not only have effect on their health and wellbeing, but also on health-seeking behaviours for their children [35]. This would enable early case detection of childhood illnesses (e.g., pneumonia and malaria) and improve child health outcomes [32]. Furthermore, emphasis should be placed on enhancing health insurance coverage, increasing the availability and accessibility of healthcare facilities, and promoting health education and literacy with respect to childhood fever [6,36,37].

**Table 2. Factors associated with delay in treatment-seeking for fever among children under five in Nigeria, 2018.**

| Variables | Crude OR (95% CI) | Adjusted OR (95% CI) |
|---|---|---|
| **Predisposing factors** | | |
| Child age | | |
| 0–11 months | 1.00 | 1.00 |
| 12–23 months | 0.86 (0.73, 1.02) | 0.84 (0.70, 1.00) |
| 24–59 months | 0.89 (0.78, 1.03) | 0.79 (0.68, 0.93) |
| Child gender | | |
| Male | 1.00 | 1.00 |
| Female | 1.04 (0.93, 1.15) | 1.02 (0.91, 1.14) |
| Mother's age | | |
| 15–24 years | 1.00 | 1.00 |
| 25–34 years | 1.04 (0.90, 1.21) | 1.19 (0.99, 1.41) |
| 35–49 years | 1.14 (0.97, 1.34) | 1.28 (1.02, 1.61) |
| Mother's education | | |
| No formal education | 1.00 | 1.00 |
| Primary education | 0.67 (0.56, 0.80) | 0.86 (0.72, 1.04) |
| Secondary or higher education | 0.47 (0.40, 0.55) | 0.88 (0.72, 1.09) |
| Father's education | | |
| No formal education | 1.00 | 1.00 |
| Primary education | 0.73 (0.61, 0.88) | 0.91 (0.75, 1.10) |
| Secondary or higher education | 0.47 (0.41, 0.56) | 0.82 (0.68, 0.99) |
| Listening to the radio | | |
| No | 1.00 | 1.00 |
| Yes | 0.56 (0.49, 0.64) | 0.85 (0.72, 0.99) |
| Reading magazine | | |
| No | 1.00 | 1.00 |
| Yes | 0.47 (0.38, 0.58) | 0.89 (0.71, 1.12) |
| Watching television | | |
| No | 1.00 | 1.00 |
| Yes | 0.52 (0.45, 0.60) | 0.93 (0.79, 1.09) |
| **Enabling factors** | | |
| Household wealth index | | |
| Poor | 1.00 | 1.00 |
| Middle | 0.60 (0.51, 0.72) | 0.83 (0.69, 1.01) |
| Rich | 0.39 (0.33, 0.46) | 0.71 (0.56, 0.89) |
| Mother's occupation | | |
| Not working | 1.00 | 1.00 |
| Formal employment | 0.70 (0.61, 0.80) | 0.84 (0.73, 0.96) |
| Informal employment | 1.28 (1.04, 1.58) | 0.95 (0.76, 1.18) |
| Distance from the health facility | | |
| Big problem | 1.00 | 1.00 |
| Not big problem | 0.51 (0.44, 0.59) | 0.88 (0.74, 1.06) |
| Mother's decisions on seeking medical care | | |
| Big problem | 1.00 | 1.00 |
| Not big problem | 0.52 (0.42, 0.64) | 0.88 (0.69, 1.12) |
| Mother's decision on money for medical care | | |
| Big problem | 1.00 | 1.00 |
| Not big problem | 0.55 (0.48, 0.63) | 0.75 (0.65, 0.87) |

(*Continued*)

**Table 2.** (Continued)

| Variables | Crude OR (95% CI) | Adjusted OR (95% CI) |
|---|---|---|
| Companion to seek medical care | | |
| Big problem | 1.00 | 1.00 |
| Not big problem | 0.56 (0.46, 0.68) | 0.97 (0.78, 1.22) |
| Need factors | | |
| Desire for the pregnancy | | |
| Desired the pregnancy | 1.00 | 1.00 |
| Not desired the pregnancy | 1.53 (1.05, 2.26) | 1.58 (1.05, 2.39) |
| Preceding birth interval | | |
| No previous birth | 1.00 | 1.00 |
| <24 months | 1.27 (1.07, 1.51) | 1.15 (0.95, 1.39) |
| 24+ months | 1.28 (1.10, 1.48) | 1.10 (0.92, 1.31) |
| Health service factors | | |
| Frequency of antenatal care use | | |
| None | 1.00 | 1.00 |
| 1–3 visits | 0.94 (0.79, 1.12) | 0.96 (0.80, 1.16) |
| 4+ visits | 0.59 (0.52, 0.66) | 0.76 (0.66, 0.88) |
| Place of birth | | |
| Home | 1.00 | 1.00 |
| Health facility | 0.58 (0.50, 0.66) | 1.07 (0.87, 1.31) |
| Community level factors | | |
| Place of residence | | |
| Urban | 1.00 | 1.00 |
| Rural | 1.91 (1.62, 2.26) | 1.17 (0.98, 1.40) |
| Geographical region | | |
| North Central | 1.00 | 1.00 |
| North East | 0.95 (0.74, 1.23) | 0.77 (0.59, 1.02) |
| North West | 0.64 (0.49, 0.82) | 0.55 (0.42, 0.73) |
| South East | 0.49 (0.37, 0.66) | 0.79 (0.58, 1.09) |
| South West | 0.50 (0.37, 0.66) | 0.84 (0.62, 1.14) |
| South South | 0.56 (0.40, 0.79) | 1.23 (0.86, 1.77) |

The Andersen Behavioural Model [29] emphasises the significance of the environment, particularly geographical regions, in analysing health service utilisation behaviours. In our study, the effect of environment is particularly more pertinent in the North-West region compared to the North-Central region due to its significantly higher rates of treatment-seeking. This could be attributed to the free health insurance scheme for vulnerable groups, aimed at improving access to health services in North-West geopolitical zone [38]. To address this regional disparity national policy will need to be modified.

## Study limitations

This study possesses several limitations. Firstly, due to the cross-sectional nature of the NDHS data, it is challenging to establish a temporal relationship between the factors examined in the study and the delay in treatment-seeking for fever, although our findings are consistent with previous studies conducted in SSA [13,28,30]. Secondly, potential confounding variables, such as the influence of social networks, which were not observed, may impact the results of our

findings. Additionally, it is important to acknowledge the potential for recall and reporting biases, as these can arise from relying on self-reported recall responses.

## Conclusions

There were significant delays in seeking treatment for childhood fever in poorer homes found in geopolitically unstable zones of Nigeria. Mothers who were poor, unemployed, and with younger children (<12 months) often delayed seeking treatment for their febrile child. Future health promotion strategies and microenterprise schemes should target both rural and urban mothers residing in poor households. Children under 12 months of age should be a priority.

## Acknowledgments

The authors are grateful to the Demographic Health Surveys Program for providing access to the Nigerian data utilised in this study.

## Author Contributions

**Conceptualization:** Anayochukwu E. Anyasodor, Kedir Y. Ahmed, Allen G. Ross.

**Data curation:** Anayochukwu E. Anyasodor, Kedir Y. Ahmed.

**Formal analysis:** Anayochukwu E. Anyasodor, Kedir Y. Ahmed.

**Methodology:** Anayochukwu E. Anyasodor, Kedir Y. Ahmed, Uchechukwu L. Osuagwu, Nnamdi C. Mgbemena, Bernd H. Kalinna, Subash Thapa, Shakeel Mahmood, Allen G. Ross.

**Supervision:** Allen G. Ross.

**Writing – original draft:** Anayochukwu E. Anyasodor.

**Writing – review & editing:** Anayochukwu E. Anyasodor, Kedir Y. Ahmed, Uchechukwu L. Osuagwu, Nnamdi C. Mgbemena, Bernd H. Kalinna, Subash Thapa, Shakeel Mahmood, Allen G. Ross.

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
