## [Decision Letter · Decision Letter 0]

7 Nov 2023

PONE-D-23-28224Delays in seeking treatment for fever in children under five years of age in Nigeria: Evidence from the National Demographic Health SurveyPLOS ONE

Dear Dr. Anyasodor,

Thank you for submitting your manuscript to PLOS ONE. After careful consideration, we feel that it has merit but does not fully meet PLOS ONE’s publication criteria as it currently stands. Therefore, we invite you to submit a revised version of the manuscript that addresses the points raised during the review process.

We look forward to receiving your revised manuscript.

Kind regards,

Sathiya Susuman Appunni, Ph D

Academic Editor

PLOS ONE

Journal Requirements:

 "N/A."

"Authors have no conflict of interest to declare."

4. We note that [Figure 1] in your submission contain [map/satellite] images which may be copyrighted. All PLOS content is published under the Creative Commons Attribution License (CC BY 4.0), which means that the manuscript, images, and Supporting Information files will be freely available online, and any third party is permitted to access, download, copy, distribute, and use these materials in any way, even commercially, with proper attribution. For these reasons, we cannot publish previously copyrighted maps or satellite images created using proprietary data, such as Google software (Google Maps, Street View, and Earth). For more information, see our copyright guidelines: http://journals.plos.org/plosone/s/licenses-and-copyright.

Natural Earth (public domain): " ext-link-type="uri" xlink:type="simple">http://www.naturalearthdata.com/"

Additional Editor Comments:

REF: PONE-D-23-28224

Dear Anayochukwu Edward Anyasodor,

Your manuscript entitled "Delays in seeking treatment for fever in children under five years of age in Nigeria: Evidence from the National Demographic Health Survey" which you submitted to Plos One, has been reviewed. The comments of the reviewers are attached to this letter. The reviewers have recommended some revisions to your manuscript. Therefore, I invite you to respond to the reviewers' comments and to revise your manuscript accordingly. Your revised manuscript should be uploaded on or before 30 November 2023.

Once the revised manuscript is prepared, you can upload it and submit it through our online submission system. Please do not upload your files as a "new submission" but as a "revised submission." You must upload two versions of your manuscript, one clean version and one with highlighted changes. To expedite the processing of the revised manuscript, please be as specific as possible in your response to the reviewers. Kindly include with your revised submission an itemized, point-by-point response to the reviewers, which details the changes made.

Feel free to contact the Editorial Office with any questions or concerns. We look forward to receiving the revised manuscript.

Sincerely,

Dr Sathiya Appunni, Ph D

PLOS ONE

Academic Editor

-------------

Reviewer 1

The manuscript is technically sound, and the data support the study's conclusion. Moreover, the statistical analysis has been performed appropriately and rigorously. Furthermore, the authors made all data underlying the findings in their manuscript fully available. I found the English language editing in this article unambiguous.

However, I suggest more reporting on bivariate analysis, where the study factors were cross-tabulated with the outcome variables in Table 1. The authors would have looked at the pattern of the percentages and reported the relationship because there is valuable information related to the topic under study.

On page 12 of the article, I was confused. Under the subheading: "Prevalence of delay in treatment-seeking for fever," (The prevalence of delay in treatment-seeking for fever was 62.1% (95%CI: 60.1, 64.1%), which was higher in children whose mothers did not attain a formal education (69.1%), and those who resided in poor households (70.0%) [Table 1]). I need help finding where the authors used the Confidence Interval in Table 1. CI was only used under the multivariate section, where they used binary logistic regression to identify the factors contributing to the predisposing, enabling, need, services, and community factors.

As it is recommended by PLOS policy, the data should be provided as part of the manuscript or its supporting information for verification.

Reviewer 2

The authors should find a basis for the choices made to declare that 1 day is a delay and, therefore, before a day, it is not a delay.

Very few recommendations came from the study. But, so many good results were obtained. It would help if you improved your recommendations.

Reviewers' comments:

Reviewer's Responses to Questions

**Comments to the Author**

1. Is the manuscript technically sound, and do the data support the conclusions?

Reviewer #1: Yes

Reviewer #2: Yes

2. Has the statistical analysis been performed appropriately and rigorously? 

Reviewer #1: Yes

Reviewer #2: Yes

3. Have the authors made all data underlying the findings in their manuscript fully available?

Reviewer #1: Yes

Reviewer #2: Yes

4. Is the manuscript presented in an intelligible fashion and written in standard English?

Reviewer #1: Yes

Reviewer #2: Yes

5. Review Comments to the Author

Reviewer #1: The manuscript technically sound, and the data support the conclusion of the study. Moreover, the statistical analysis has been performed appropriately and rigorously. Furthermore, the authors made all data underlying the findings in their manuscript fully available. I found the English language editing in this article clear and unambiguous.

However, i would suggest more re porting on bivariate analysis, where the study factors were crosstabulated with the outcome variables in table 1. The authors would have looked at the pattern of the percentages and report the relationship because there is a useful information related to the topic under study.

On page 12 of the article, i was a bit confused. Under the subheading: "Prevalence of delay in treatment-seeking for fever" (The prevalence of delay in treatment-seeking for fever was 62.1% (95%CI: 60.1, 64.1%), which was higher in children whose mothers did not attain a formal education (69.1%), and those who resided in poor households (70.0%) [Table 1]). I fail to see where the authors used Confidence Interval in table 1. CI was only used under multivariate section, where they used binary logistic regression to identify the factors contributing to the predisposing, enabling, need, services, and community factors.

As it is recommended by PLOS policy, I would suggest The data to be provided as part of the manuscript or its supporting information and for verification.

Reviewer #2: The authors should find a basis for the choices made to declare that 1 day is a delay and therefore before a day it is not a delay.

Very few recommendations came from the study. But, so many good results were obtained. You need to improve your recommendations.

6. PLOS authors have the option to publish the peer review history of their article (what does this mean?). If published, this will include your full peer review and any attached files.

Reviewer #1: No

Reviewer #2: No

---

## [Author Response · Author response to Decision Letter 0]

23 Nov 2023

Academic Editor

Comment: Please ensure that your manuscript meets PLOS ONE's style requirements, including those for file naming. The PLOS ONE style templates can be found at https://journals.plos.org/plosone/s/file?id=wjVg/PLOSOne_formatting_sample_main_body.pdf and https://journals.plos.org/plosone/s/file?id=ba62/PLOSOne_formatting_sample_title_authors_affiliations.pdf

Response: The manuscript has been revised to meet PLOS ONE's style requirements.

Comment: Thank you for stating the following financial disclosure: "N/A. At this time, please address the following queries: a) Please clarify the sources of funding (financial or material support) for your study. List the grants or organizations that supported your study, including funding received from your institution. b) State what role the funders took in the study. If the funders had no role in your study, please state: “The funders had no role in study design, data collection and analysis, decision to publish, or preparation of the manuscript.” c) If any authors received a salary from any of your funders, please state which authors and which funders. d) If you did not receive any funding for this study, please state: “The authors received no specific funding for this work.”

Response: We have edited this in the revised manuscript (Page 15, line 303). The financial disclosure statement reads: “The authors received no specific funding for this work.”

Comment: Thank you for stating the following in your Competing Interests section: "Authors have no conflict of interest to declare." Please complete your Competing Interests on the online submission form to state any Competing Interests. If you have no competing interests, please state "The authors have declared that no competing interests exist.", as detailed online in our guide for authors at http://journals.plos.org/plosone/s/submit-now This information should be included in your cover letter; we will change the online submission form on your behalf.

Response: We have edited this in the revised manuscript (Page 15, line 304). The competing interests statement reads: “The authors have declared that no competing interests exist.”

Comment: We note that [Figure 1] in your submission contain [map/satellite] images which may be copyrighted. All PLOS content is published under the Creative Commons Attribution License (CC BY 4.0), which means that the manuscript, images, and Supporting Information files will be freely available online, and any third party is permitted to access, download, copy, distribute, and use these materials in any way, even commercially, with proper attribution. For these reasons, we cannot publish previously copyrighted maps or satellite images created using proprietary data, such as Google software (Google Maps, Street View, and Earth). For more information, see our copyright guidelines: http://journals.plos.org/plosone/s/licenses-and-copyright. We require you to either (1) present written permission from the copyright holder to publish these figures specifically under the CC BY 4.0 license, or (2) remove the figures from your submission:

Response: We have now removed Figure 1 from the revised manuscript.

Comment: Please review your reference list to ensure that it is complete and correct. If you have cited papers that have been retracted, please include the rationale for doing so in the manuscript text, or remove these references and replace them with relevant current references. Any changes to the reference list should be mentioned in the rebuttal letter that accompanies your revised manuscript. If you need to cite a retracted article, indicate the article’s retracted status in the References list and also include a citation and full reference for the retraction notice.

Response: We have edited the reference in the revised manuscript.

Reviewer #1

Comment: The manuscript is technically sound, and the data support the study's conclusion. Moreover, the statistical analysis has been performed appropriately and rigorously. Furthermore, the authors made all data underlying the findings in their manuscript fully available. I found the English language editing in this article unambiguous. 

Response: We thank the reviewer for the comments. The reviewer’s specific comments are addressed below in this rebuttal.

Comment: However, I suggest more reporting on bivariate analysis, where the study factors were cross-tabulated with the outcome variables in Table 1. The authors would have looked at the pattern of the percentages and reported the relationship because there is valuable information related to the topic under study. 

Response: We have edited this in the revised manuscript (Table 1, appended after references).

Comment: On page 12 of the article, I was confused. Under the subheading: "Prevalence of delay in treatment seeking for fever," (The prevalence of delay in treatment-seeking for fever was 62.1% (95%CI: 60.1, 64.1%), which was higher in children whose mothers did not attain a formal education (69.1%), and those who resided in poor households (70.0%) [Table 1]). I need help finding where the authors used the Confidence Interval in Table 1. CI was only used under the multivariate section, where they used binary logistic regression to identify the factors contributing to the predisposing, enabling, need, services, and community factors. 

Response: We have edited this in the revised manuscript (Page 11, lines 218-219)

Comment: As it is recommended by PLOS policy, the data should be provided as part of the manuscript or its supporting information for verification. 

Response: We acknowledge the reviewer's concern and have addressed it by including a data-sharing statement in the revised manuscript (Page 15, lines 313-318): The data used for this study are available from The DHS Program. Users can register on the DHS Program website (https://dhsprogram.com/). Once registered, interested researchers can request access to the datasets. The 2018 Nigeria DHS data are available for download at the following link: https://dhsprogram.com/data/dataset/Nigeria_Standard-DHS_2018.cfm?flag=0. The authors confirm they had no special access privileges.

Reviewer #2

Comment: The authors should find a basis for the choices made to declare that 1 day is a delay and, therefore, before a day, it is not a delay.

Response: We selected 24hrs for defining a delay in seeking care for fever based on information from the Guide to DHS Statistics (page 12.24, available at https://www.dhsprogram.com/pubs/pdf/DHSG1/Guide_to_DHS_Statistics_DHS-7_v2.pdf) and previously published studies (Bekele et al., 2023; Debsarma et al., 2022; Hamooya et al., 2016; Rutebemberwa et al., 2009). This is now reflected in the revised manuscript (Page 7 lines 142-145)

Comment: Very few recommendations came from the study. But, so many good results were obtained. It would help if you improved your recommendations.

Response: The ‘Conclusion’ has been revised (Page 14, lines 294-299).

---

## [Editor Report · Decision Letter 1]

29 Nov 2023

Delays in seeking treatment for fever in children under five years of age in Nigeria: Evidence from the National Demographic Health Survey

PONE-D-23-28224R1

Dear Dr. Anayochukwu,

We’re pleased to inform you that your manuscript has been judged scientifically suitable for publication and will be formally accepted for publication once it meets all outstanding technical requirements.

Kind regards,

Sathiya Susuman Appunni, Ph D

Academic Editor

PLOS ONE
---

## [Editor Report · Acceptance letter]

11 Dec 2023

PONE-D-23-28224R1 

Delays in seeking treatment for fever in children under five years of age in Nigeria: evidence from the National Demographic Health Survey 

Dear Dr. Anyasodor:

I'm pleased to inform you that your manuscript has been deemed suitable for publication in PLOS ONE. Congratulations! Your manuscript is now with our production department. 

Kind regards, 

on behalf of

Professor Sathiya Susuman Appunni 

Academic Editor

PLOS ONE